# Flying squirrels use a mortise-tenon structure to fix nuts on understory twigs

Han Xu[1,2†], Lian Xia[2†], John R Spence[3†], Mingxian Lin[1], Chunyang Lu[1,2], Yanpeng Li[1], Jie Chen[1], Tushou Luo[1], Yide Li[1], Suqin Fang[4]*

[1]Research Institute of Tropical Forestry, Chinese Academy of Forestry, Guangzhou, China; [2]College of Forestry, Hainan University, Haikou, China; [3]Department of Renewable Resources, University of Alberta, Edmonton, Canada; [4]State Key Laboratory of Biocontrol, School of Life Sciences, Sun Yat-Sen University, Guangzhou, China

**Abstract** Squirrels of temperate zones commonly store nuts or seeds under leaf litter, in hollow logs, or even in holes in the ground; however, in the humid rainforests of Jianfengling in Hainan, South China, we show that some flying squirrels cache elliptical or oblate nuts by hanging them securely in vegetation. These small flying squirrels were identified as *Hylopetes phayrei electilis* (G. M. Allen, 1925) and *Hylopetes alboniger* (Hodgson, 1870), in video clips captured of their behavior around focal nuts. Squirrels chewed grooves encircling ellipsoid nuts or distributed on the bottoms of oblate nuts, and then used these grooves to fix nuts tightly between small twigs 0.1–0.6 cm in diameter that were connected at angles of 25–40°. The grooves carved on the nuts (concave structure) connected with Y-shaped twigs (convex structure) and thus firmly affixed the nuts to the plant in a way similar to a mortise-tenon joint used in architecture and carpentry. Cache sites were on small plants located 10–25 m away from the closest potentially nut-producing tree, a behavior that likely reduces the discovery and consumption of the nuts by other animals. The adaptive squirrel behavior that shapes and fits nuts between twigs seems to be directed at providing more secure storage that increases food supply during dry periods in a humid tropical rainforest. In addition to providing such benefits for the squirrels, we suggest that this behavior also impacts the distribution of tree species in the forest.

## Editor's evaluation

This report of nut modification and storage in flying squirrels provides new insights into food caching behaviour in wild animals. Although further direct evidence is needed to corroborate some of the findings, the current study provides valuable documentation of an interesting behaviour that should motivate further observational and experimental research.

*For correspondence: fangsuq5@mail.sysu.edu.cn

†These authors contributed equally to this work

Competing interest: The authors declare that no competing interests exist.

## Introduction

Storing food to buffer against periods of low resource supply is a common species-specific behavior used by squirrels and other rodents (*Andersson and Krebs, 1978*; *Steele et al., 2006*). Nuts, in particular, are harvested from trees and cached in various places. For example, many temperate-zone squirrels hoard nuts under leaf litter, in holes in trees or logs or in the ground (*Cheng et al., 2005*; *Hadj-chikh et al., 1996*). In subtropical zones, however, some species store nuts or mushrooms by hanging them on tree branches, a behavior thought to minimize fungal infection in humid environments (*Lichti et al., 2017*; *Xiao et al., 2013*) or decrease the risk of loss through decomposition or germination under warmer temperatures in the cache (*Sechley et al., 2015*).

**eLife digest** The rainy forests of South China are home to *Cyclobalanopsis* trees whose smooth, elliptical nuts are favoured by many animal species. While doing fieldwork in the Jianfengling nature reserve in the southern province of Hainan, China, researchers came across an unusual sight: many of these nuts had been wedged into the Y-shaped forks between diverging twigs. A closer inspection revealed that a carefully crafted groove on the surface of the nuts helped them to stay wedged and secured between the branches. Which creature was responsible for such a feat?

To investigate, Xu et al. set up motion-triggered, infra-red cameras near some of the hoarding sites. They discovered that the culprits were *Hylopetes phayrei electilis* and *Hylopetes alboniger*, two small species of flying squirrel that tend to store *Cyclobalanopsis* nuts to prepare for the dry, cool season.

The footage showed that the squirrels first chewed the nuts before inserting them tightly between the branches. In fact, this process appeared to require much care – and, potentially, cognitive involvement – with the squirrels testing and adjusting their grooves many times until a perfect fit was achieved. Caching sites were usually found 10 to 25 meters away from the nearest *Cyclobalanopsis* tree, which probably helps to protect the hoards from other animals on the hunt for nuts.

Squirrels from temperate regions typically prepare for winter by hiding food in the ground, between logs or inside hollow trees; in humid, tropical forests, however, such caching sites may promote mould, decomposition or germination. In these conditions, securely hanging nuts between branches may prove to be a more suitable strategy. By choosing caching sites that are away from the mother tree, squirrels may also inadvertently help *Cyclobalanopsis* to expand their range, with forgotten nuts becoming dislodged and sprouting in new locations across the reserve. Overall, these findings shed new light on animal adaptation and cognition, as well as on the forces that help to shape forest ecology.

The present work was prompted by our inadvertent discovery of *Cyclobalanopsis* nuts with strange surface grooves, and that were suspended in Y-shaped crotches of twigs on understory plants on Hainan Island, South China. *Cyclobalanopsis* trees are dominant fagaceous trees in these tropical forests; however, their fruits are elliptical or oblate single nuts with smooth surfaces, features that make them difficult to hang on vegetation. Firmly suspending such nuts in the vegetation presents an ecological challenge to squirrels in such environments. We asked whether some of the nine squirrel species identified from Hainan forests used special behaviors to prepare these nuts and fix them securely on vegetation.

In this paper, we show that the Indochinese Flying Squirrel, *Hylopetes phayrei electilis* (G. M. Allen, 1925), and the Particolored Flying Squirrel, *H. alboniger* (Hodgson, 1870), which co-occur in Hainan Island, cache these nuts individually between the twigs of small plants. Both of these small-bodied flying squirrels are widespread in the tropical forests in Southeast Asia, from Myanmar, south to northwestern Vietnam, and east into southern China (*Duckworth et al., 2016*; *Duckworth et al., 2016*). In China, *H. phayrei electilis* can be found in the mountainous areas of Hainan, Fujian, Guangxi, and Guizhou Provinces. *H. alboniger* is mainly found in the provinces of Hainan, Yunnan, Guizhou, Guangxi and, more rarely, in Zhejiang.

Although these squirrels are reasonably common, there is little published information about their habits, and there are no studies from Hainan Province in China (*Li et al., 2012*). In particular, nut storage behavior hasn't been reported from elsewhere in the ranges of either of these two squirrels. Thus, the preparation of nuts to connect them firmly to twigs is a new finding, although other squirrel species are known to handle nuts prior to suspending them to improve the success rate of storage (*Fox, 1982*; *Steele and Yi, 2020*; *Xiao et al., 2010*). Here, we document in some detail the squirrel behaviors associated with this phenomenon in Hainan.

## Results and discussion

We used images from infrared cameras to determine that the nocturnal flying squirrels, *H. phayrei electilis* and *H. alboniger,* two of the nine species of squirrels known from tropical forests of Hainan

**Table 1.** The nine squirrel species known from Jianfengling, Hainan Island, China.

| Species and subspecies name | Body length/mm |
| --- | --- |
| *Tamiops maritimus* (Bonhote,1900) (*Liu et al., 2020*; *Pan et al., 2007*) | 105～134 |
| *Dremomys pyrrhomerus* (Thomas, 1895) (*Xu and Chen, 1989*) | 194～215 |
| *Hylopetes alboniger* (Hodgson, 1870) (*Liu et al., 2020*; *Andrew, 2008*; *Pan et al., 2007*) | 180～203 |
| *Hylopetes phayrei electilis* (Allen, 1925) (*Liu et al., 2020*; *Pan et al., 2007*) | 123～173 |
| *Dremomys rufigenis* (Blanford, 1878) (*Zheng et al., 2008*) | 170～250 |
| *Callosciurus erythraeus* (Pallas, 1779) (*Huang, 1995*; *Zheng et al., 2008*) | 198～252 |
| *Belomys pearsonii* (Gray, 1842) (*Pan et al., 2007*; *Huang, 1995*) | 180～260 |
| *Petaurista albiventer* (Gray, 1834) (*Jing et al., 2007*) | 420～520 |
| *Ratufa bicolor* (Sparrmann, 1778) (*Li et al., 2008*) | 350～505 |

Note: The data in this table are referenced from the below literature.

(*Table 1*), stored *Cyclobalanopsis* nuts by suspending them on vegetation in the Jianfengling forest (*Videos 1–5*). The videos further showed that the squirrels chewed grooves in the surfaces of the nuts before fixing between the twigs, and that they sometimes altered the previously carved grooves by further chewing, apparently to adjust the fit and suspend the nut more firmly (*Videos 6–9*). In footage from 32 field infrared cameras, we captured 48 film sequences that included chewing (*Videos 6–7*), fixing (*Video 9*, partial evidence) and removing nuts (*Video 2*, *Video 3* and *Video 5*), or visiting a storage site (*Video 1*, *Video 4* and *Video 8*). This direct evidence, together with the findings below, shows that this mode of nut storage is a reasonably common activity of these two squirrel species in the Jianfengling forest.

A total of 151 grooved and cached nuts were found suspended on more than 55 tree or shrub species distributed across 28 plant families during our censuses of approximately 5.5 ha of forest (*Figure 1*, *Supplementary file 1*). All suspended nuts found had surface grooves of the form carved by squirrels, as documented above. Examples of storage locations and carved nuts are shown in *Figures 2–3*. Most discovered nuts were fixed between plant twigs connected at angles of 25–40° on a variety of small saplings and shrubs (*Figure 4*). This range of angles accommodates the nut sizes of *Cyclobalanopsis edithiae* and *C. patelliformis* (2.4 cm (width) × 4.6 cm (length) and 2.4 cm (width) × 2.0 cm (height), respectively), which accounted for 96.7% of the nuts that we found cached (*C. edithiae* (40.4%), *C. patelliformis* (56.3%)). A few nuts of *Lithocarpus fenzelianus* A. Camus (n=4) and *C. fleuryi* (Hickel & A. Camus) Chun ex Q. F. Zheng (n=2) were also found similarly suspended on plants.

Nuts of the two predominant tree species were disproportionately stored on small plants with diameters at breast height (DBH) of 0.4–1.6 cm (*Figure 5*) and twig diameters of 0.10–0.60 cm (*Figure 6a and b*). For nuts of *C. edithiae*, plant twig diameter was significantly correlated with groove width on the nut, and generally varied from 0.20–0.60 cm (p<0.001, *Figure 7*). The widths of grooves carved in

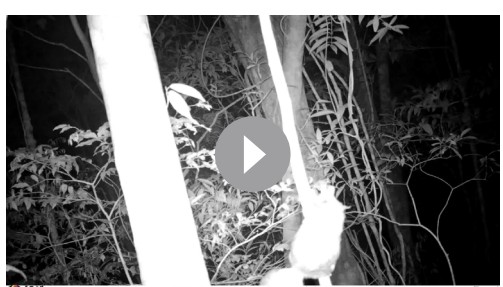

**Video 1.** Squirrel 1 of *Hylopetes alboniger* was checking and re-fixing nuts at the storage sites with footage from infrared cameras.

https://elifesciences.org/articles/84967/figures#video1

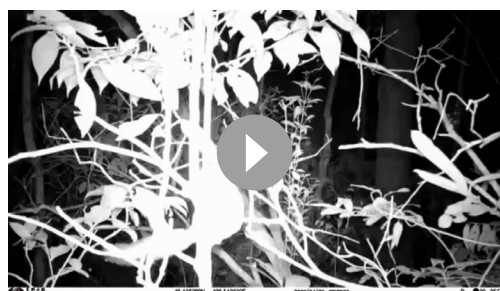

**Video 2.** Squirrel 2 of *Hylopetes alboniger* was removing nuts from storage sites with footage from infrared cameras.

https://elifesciences.org/articles/84967/figures#video2

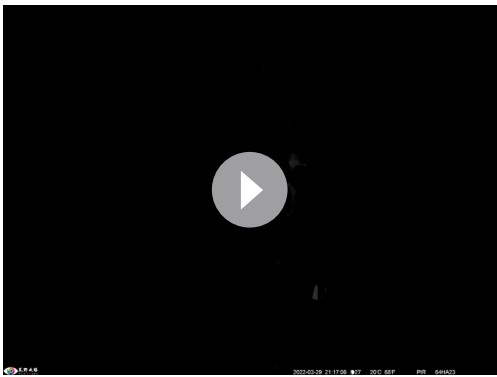

**Video 3.** Squirrel 3 of *Hylopetes alboniger* was removing nuts from storage sites with footage from infrared cameras.

https://elifesciences.org/articles/84967/figures#video3

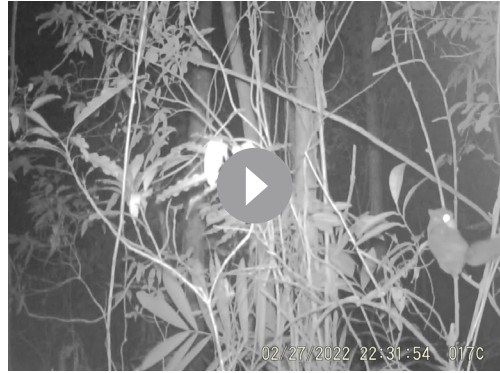

**Video 4.** Squirrel 4 of *Hylopetes phayrei electili* was checking and re-fixing nuts at storage sites with footage from infrared cameras.

https://elifesciences.org/articles/84967/figures#video4

the nuts matched the typical width of the paired incisor tips of these squirrels (i.e. less than 5 mm). Most of the nuts were found stored on the first to third branches of a plant 1.50–2.50 m above the ground (45.9% of *C. edithiae* and 43.5% of *C. patelliformis* storage sites) (*Figure 8*).

Squirrels of both species carved spiral zigzagged grooves that encircled the midsection surface of the ellipsoid nuts of *C. edithiae* (*Figure 2a–f*) with one, or occasionally, two grooves (*Figure 3a–c*). Two non-connected or spiral grooves appeared to be useful for adjusting the position of stored nuts to the specific orientation of the twigs. In contrast, up to 20 surface grooves were carved on the bottoms of the oblate nuts of *C. patelliformis* (*Figure 2g–i*). These grooves on oblate nuts varied considerably in pattern from symmetric (*Figure 3d–i*) to scattered. Symmetrical grooves on the bottom of nuts likely facilitate firm positioning as squirrels rotated nuts, apparently to optimize the nut's position for the most secure attachment.

Interestingly, oblate nuts stored on living trees and shrubs had significantly more carved shallow scattered grooves than those stored on dead trees and lianas (5.1 ± 5.0 *vs* 2.8 ± 4.0, t=2.1591, df=46.402, p=0.036). Because the bark of dead trees and lianas is coarser than that of living trees, fewer grooves may be required to hold the nuts securely in place. We also note that the grooves on the ellipsoid nuts of *C. edithiae* were deeper (more than 0.5 mm) than those on the oblate nuts of *C. patelliformis* (less than 0.45 mm, p<0.05, *Figure 9*). Nonetheless, none of the chewed grooves that we observed were deep enough to damage the endosperm of the nut, and thus the squirrels seemed to minimize the potential impacts of fungi during storage.

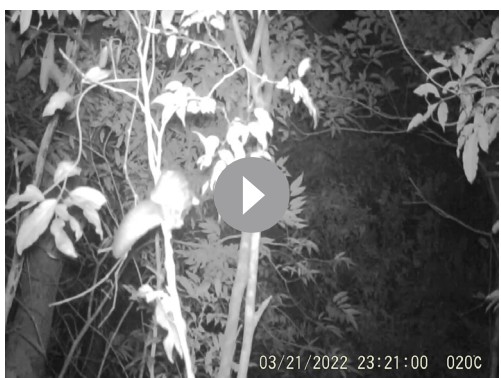

**Video 5.** Squirrel 5 of *Hylopetes phayrei electili* was removing nuts from storage sites with footage from infrared cameras.

https://elifesciences.org/articles/84967/figures#video5

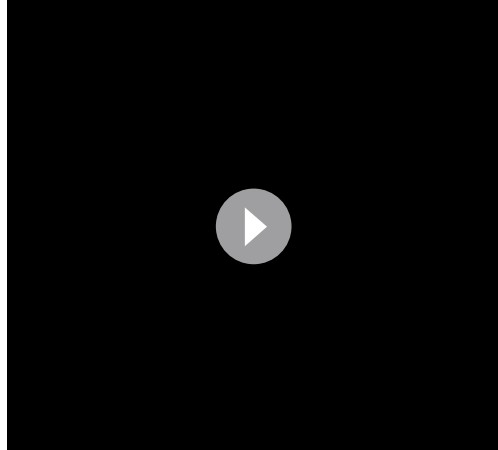

**Video 6.** One squirrel was cracking the nuts on the ground with footage from infrared cameras.

https://elifesciences.org/articles/84967/figures#video6

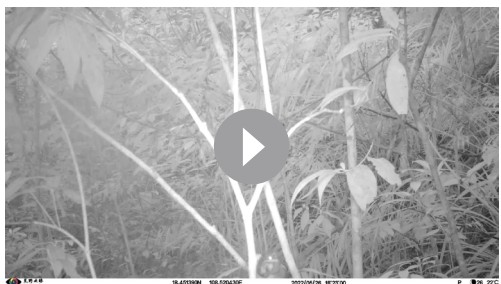

**Video 7.** One squirrel was rotating and cracking the nuts on the trees with footage from infrared cameras. https://elifesciences.org/articles/84967/figures#video7

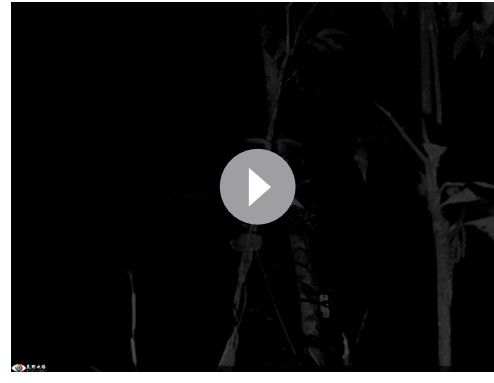

**Video 8.** One squirrel was re-fixing the nuts by cracking behavior with footage from infrared cameras. https://elifesciences.org/articles/84967/figures#video8

The surface grooves allowed the squirrels to 'pressure fit' the nuts between the two plant twigs in a way functionally similar to a mortise-tenon joint (*Qiao et al., 2021*; *Figure 3*). Squirrels used the twigs as a convex 'tenon' to fit into the convex 'mortise' provided by the grooves on the nuts. Thus, carved nuts were inlayed between plant twigs (0.10–0.60 cm in diameter) intersecting at specific angles (25–40°) on various understory plants (*Figure 4*). We found nuts on small trees and shrubs, but also on lianas, bamboos, or dead trees, and even occasionally on large petioles of palms or trees (*Figure 2j–n*, *Table 2*). Once fixed in this manner, nuts were resistant to being blown off by strong wind or even by shaking that we administered experimentally (*Videos 10–15*).

The distance between the closest *Cyclobalanopsis* trees producing nuts and storage sites on smaller understory plants varied from 10–25 m (*Figure 10*), distances greater than the average canopy width of large trees in the Jianfengling forest (estimated to be 10 ± 5 m). This sort of distancing likely reduces discovery by other squirrels, mice, or other animals potentially searching for aboveground nuts below the parent trees (*Cao et al., 2011*), although recordings from our cameras show that some nuts were still found and eaten by mice.

Because of this spacing, most seedlings that result from dropped, fallen or forgotten nuts (*Figure 11*) will germinate at some distance from their parents. Thus, seed dispersal by these squirrels may decrease competition between seedlings and parent trees. This should increase seedling survival rates and could, in turn, truly decrease the negative density dependence of conspecific trees (see *Detto et al., 2019*). Unfortunately, we presently do not have sufficient data to estimate what proportion of the disappearance that we observed is the result of use by squirrels, although our video footage establishes that they do remove some nuts (*Videos 2–5*). Nonetheless, some proportion of nuts likely falls from storage sites and germinates nearby, as is common for seeds and nuts cached by squirrels, especially in hardwood forests (*Steele and Yi, 2020*).

Only 63.6% of nuts that we discovered on understory plants were fresh at the time of the survey. Under natural conditions, these nuts on the ground would likely germinate in ca. two to three months after they had fallen to the ground (*Zhou, 2001*). These stored nuts did not germinate during our 3.5 month investigation interval, which means that nuts can persist in these storage sites for longer periods than do nuts on the ground. Over the 44 days between the first and second surveys, 19.7% of the stored nuts had disappeared, and 15.0% of the nuts discovered during the second survey were new. Over the 61 days between the second and third surveys, 43.7% of the stored nuts had disappeared, and

On Jan. 23,2023

*Hylopetes phayreie* was ste... *Cyclobalanopsis edithiae* between ... of a tree captured by infrared camera in Jianfengling tropical rainforest.

**Video 9.** One squirrel was fixing the nuts between the twigs with footage from infrared cameras. We merged several photos and a video successively taken by an infrared camera in 30 s. https://elifesciences.org/articles/84967/figures#video9

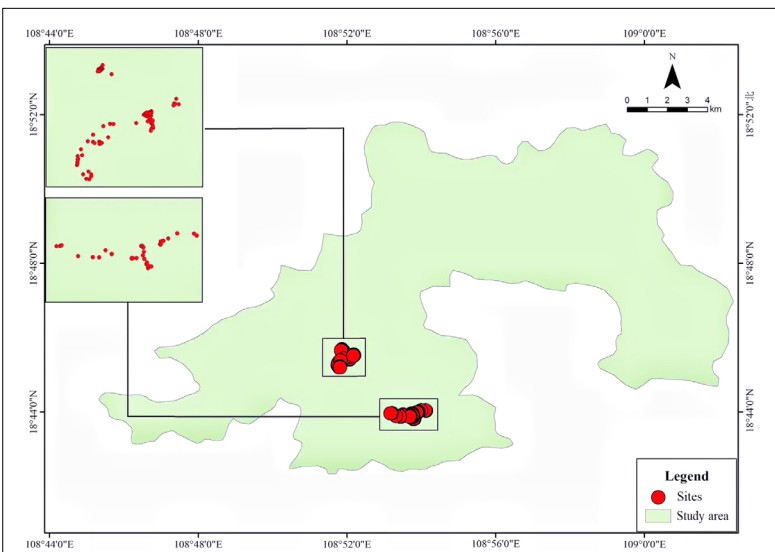

**Figure 1.** Spatial distribution of the 151 suspended nuts observed in Jianfengling Nature Reserve, Hainan, China.

20.6% of the nuts discovered were new. Thus, the numbers and composition of stored nuts in this forest are dynamic variables.

In general, the number of nuts being stored decreased gradually from January to May after the fruiting season, i.e., it decreased gradually through the dry season toward the rainy season. The high precipitation and humidity of the Jianfengling forest environment (*Xu et al., 2015*) likely favors the

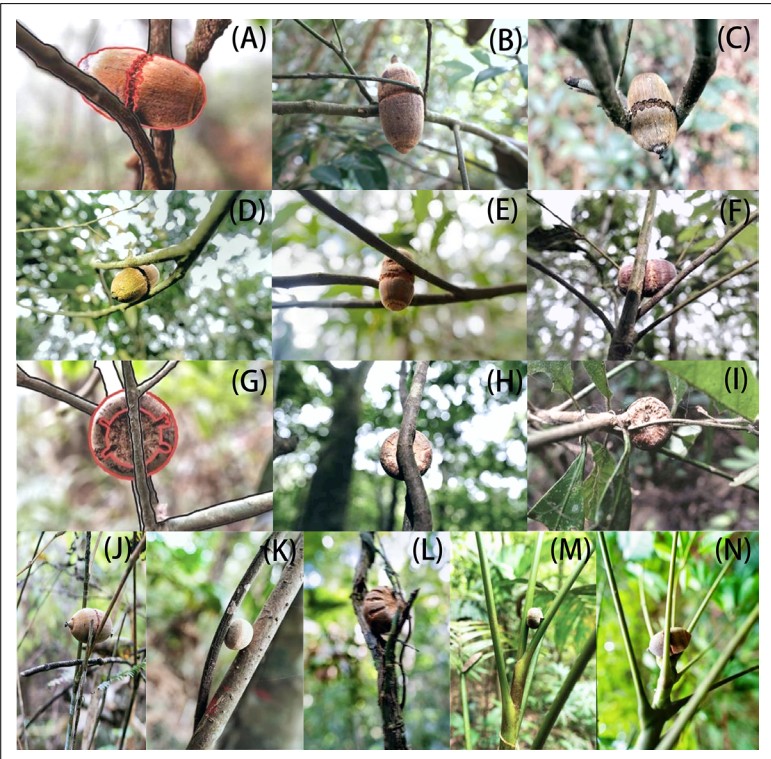

**Figure 2.** Nuts are stored after surface preparation by flying squirrels. (**A**) Nut of *C. edithiae* (Skan) Schottky, with chewed grooves outlined in red. Nuts of *C. edithiae* fixed on trees, with (**B–D**) one groove, (**E**) two non-connected grooves, or (**F**) spiral carved grooves encircling the nuts. (**G**) Nut of *C. patelliformis* (Chun) Y. C. Hsu et H. W. Jen, with chewed grooves outlined in red. (**H–I**). Nuts of *C. patelliformis* fixed on trees, with carved grooves on the bottom fixed on (**J**) bamboos, (**K–L**) lianas, between the big petioles of (**M**) trees and (**N**) palms.

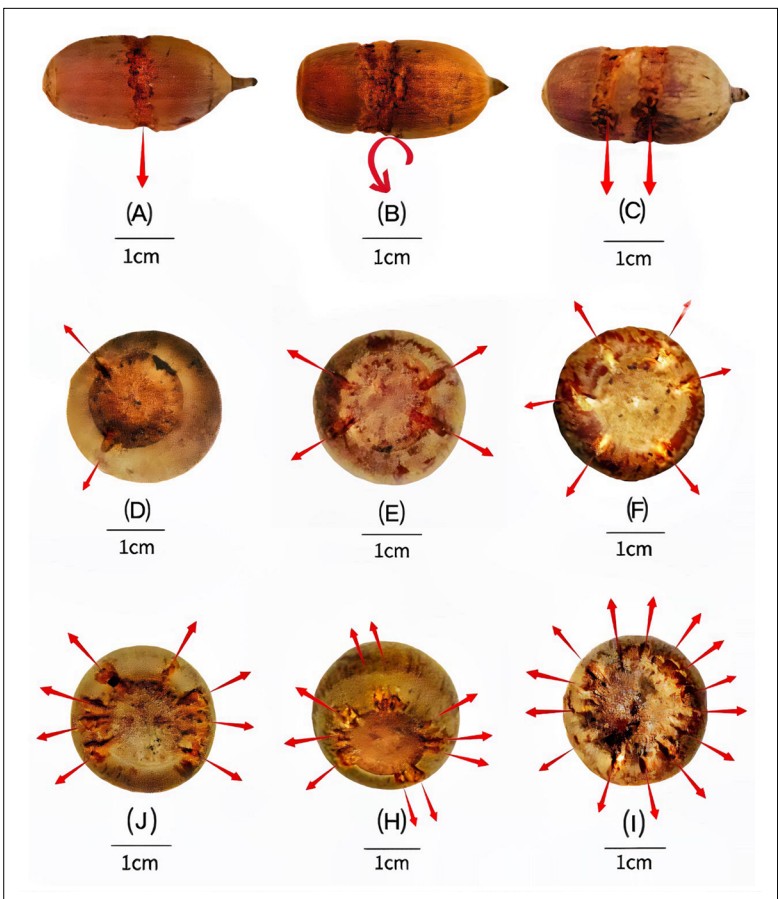

**Figure 3.** Variation in carved grooves depends on the storage situation. The carved surface grooves on nuts of *C. edithiae* mostly encircle the middle of the nut, with (**A**) one groove, (**B**) one spiral groove, or (**C**) two separated grooves. The grooves on nuts of *C. patelliformis* are distributed on the bottom of the nuts, with (**D**) 2, (**E**) 4, (**F**) 6, (**G**) 8, (**H**) 10 symmetrically, or (**I**) randomly distributed grooves.

storage of nuts above the ground by reducing fungal infection or germination and may influence the timing of removal of stored nuts. In more temperate forests with lower annual precipitation nuts can be safely stored under dry leaf litter or in the ground without special processing (*Hadj-chikh et al., 1996*). Hence, the suspended storage that we observed appears to be an effective adaptation for safe storage, mainly during drier seasons to improve the food supply for the squirrels during the colder

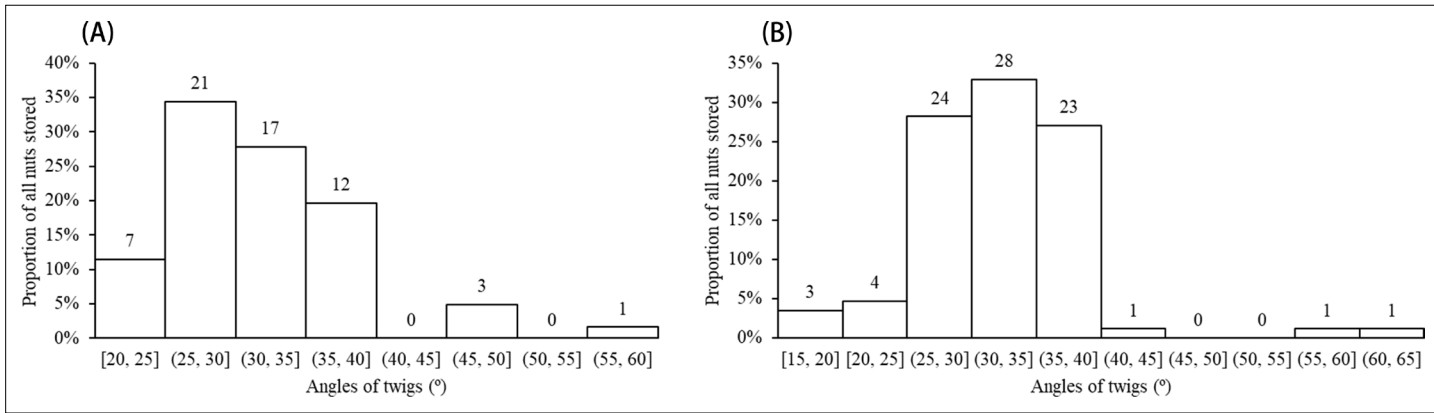

**Figure 4.** Nuts were fixed tightly between twigs generally meeting at angles of 25–40°. (**A**) *C. edithiae* nuts. (**B**) *C. patelliformis* nuts.

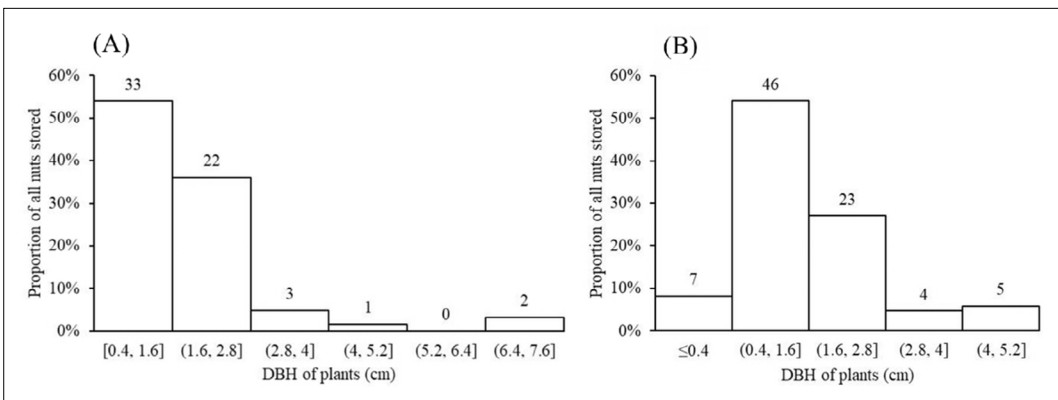

**Figure 5.** Most nuts were stored on small plants with the diameter at breast height (DBH) ranging from 0.4–1.6 cm. (**A**) *C. edithiae* nuts. (**B**) *C. patelliformis* nuts. Notes: The value on each bar is the actual number of stored nuts.

months at Jianfengling. However, field comparative data about the fates of seeds stored on or above are needed to properly evaluate this hypothesis.

In summary, we have demonstrated that individuals of the flying squirrel species *H. phayrei electilis* and *H. alboniger* collect and cache nuts from or beneath two species of *Cyclobalanopsis* trees in the Jianfengling forest. Nuts were carried 10–25 m away from parental fruiting trees and processed for storage by chewing grooves into their surfaces before they were suspended on shrubs or small trees (*Videos 6–7*). The pattern and depth of these grooves varied with the shape of nuts from these two *Cyclobalanopsis* trees so that they could be effectively fixed in the crotches of two twigs. Squirrels appear to check the strength of fixation, and sometimes iteratively modify the grooves to improve the attachment (*Video 1*, *Video 4* and *Video 8*) before a nut is finally removed from a storage site (*Video 2*, *Video 3* and *Video 5*).

Clearly, individuals of these two squirrels store nuts of different shapes and sizes securely on a variety of plant twigs at some distance from the plants that produced the nuts. The behavior of these two squirrel species has evolved to prepare nuts for such storage by chewing surficial grooves on nuts to enable a 'mortise-tenon' connection between nuts and understory plant twigs. The significant

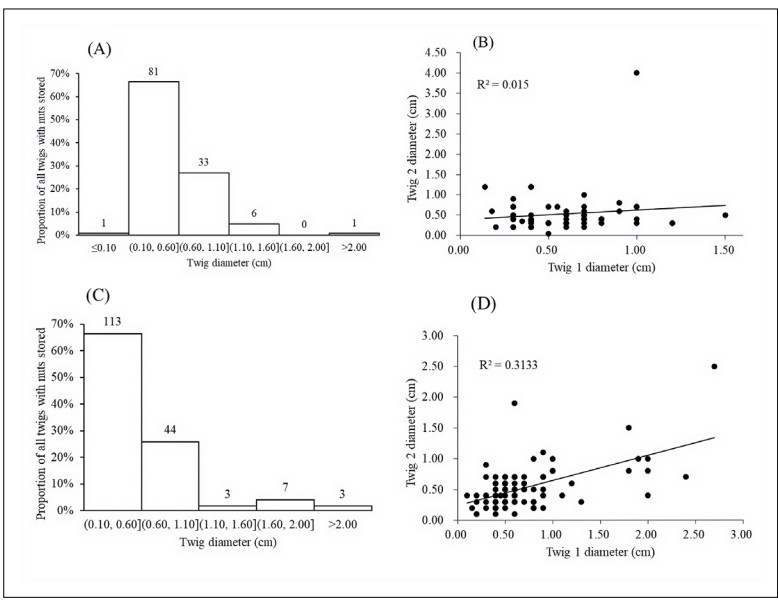

**Figure 6.** Nuts were stored mainly on small plants between twigs with diameters of 0.10 - 0.60 cm. (**A**) Histogram of diameters of twigs used to store nuts of *C. edithiae*. (**B**) Histogram of diameters of twigs used to store nuts of *C. patelliformis*. Notes: The value on each bar is the actual twigs with the number of stored nuts.

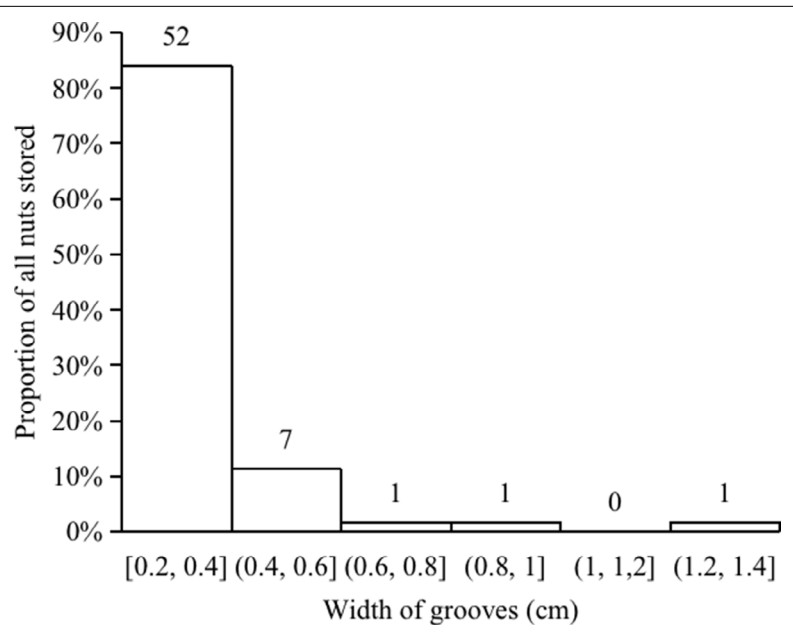

**Figure 7.** Grooves carved by squirrels on most *C. edithiae nuts* were 0.2 - 0.6 cm in width.

efforts that we observed of squirrels testing and adjusting the fixation of nuts suggest that they employ active cognitive processes in storing these nuts.

Taken together, our observations suggest that effective food storage behavior is a significant aspect of the adaptation of these two flying squirrel species to life in the humid tropical rainforest. Nut caching behavior helps to secure food for the coolest month in these rainforests. We have not yet compared the fates of suspended nuts with those of the same species buried in the ground or leaf litter, but predict that suspended storage will be far superior. The caching behavior may further affect the dispersal of nuts (*Chang and Zhang, 2014*; *Xiao et al., 2004*) in a way that alters the spatial and temporal distribution of the local plant community in the long run. Thus, this behavior could have a significant impact on the larger forest community. Although the importance of large DBH trees has been emphasized for the maintenance of forest ecosystem productivity (*Lutz et al., 2018*); however, from a broader perspective that includes the understanding of squirrel caching behavior, small understory plants may help sustain the diversity and complexity of forest structure. The possibility that such plant-animal interactions affect tree populations and distributions deserves more attention in the future (*Goheen and Swihart, 2003*; *Rong et al., 2013*).

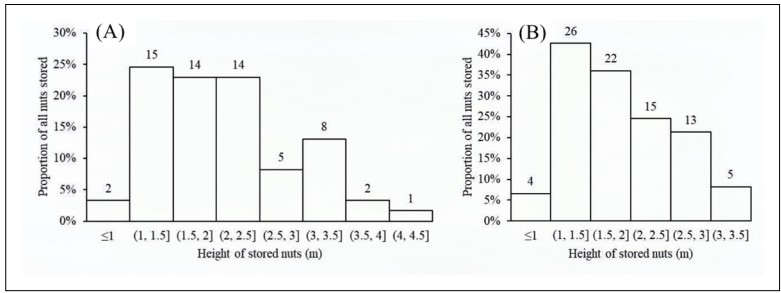

**Figure 8.** Nuts were generally stored on the first to third branches at 1.5–2.5 m aboveground. (**A**) *C. edithiae* nuts. (**B**) *C. patelliformis* nuts. Notes: The value on each bar is the actual number of stored nuts.

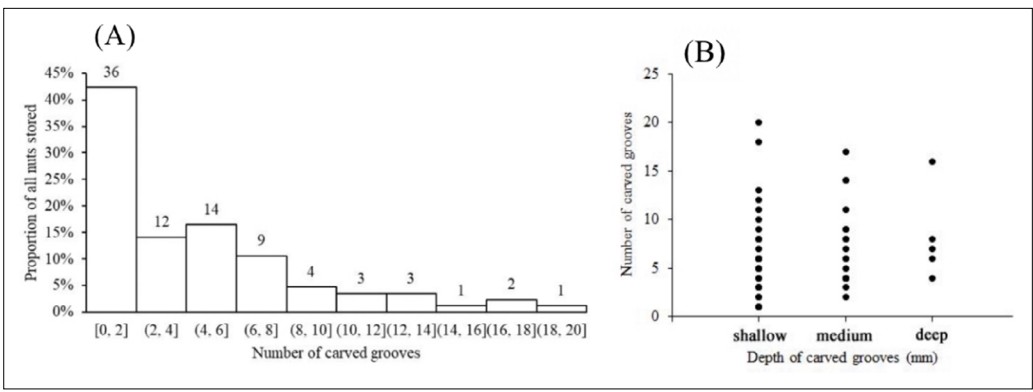

**Figure 9.** Number of grooves carved on the oblate nuts of *C. patelliformis*. (**A**) Most nuts had fewer than eight grooves. (**B**) The depth of most grooves was shallow to medium. Notes: The value on each bar is the actual number of stored nuts.

## Materials and methods
### Study site
This study was conducted in the Jianfengling region of the Hainan Tropical Rainforest National Park in Hainan Province, China (108°46'–109°45'E). The area has a seasonal tropical monsoon climate with a rainy season from June to October and a dry season from November through May of the next year. The mean annual temperature in this forest is 19.7 °C and the annual average precipitation is 2461 ± 619 mm. Jianfengling is the second rainiest area on Hainan Island, with an average annual relative humidity of >88% (*Jiang and Lu, 1991*).

The Jianfengling forest includes 992 freestanding tree and shrub species, and is dominated by trees of Fagaceae, Lauraceae, and Moraceae (*Xu et al., 2012*). *Castanopsis, Lithocarpus,* and *Cyclobalanopsis* are the three main genera of Fagaceae, which reproduce through nuts that are used as food by various mammals. The cupules of *Castanopsis* are solitary units produced on a rachis, completely or partially enclosing the nut, while cupules of *Lithocarpus* are grouped together in cymes on the rachis, completely or partly enclosing the nut. In contrast, cupules of *Cyclobalanopsis* are solitary, and do not enclose the nuts. Because enclosed nuts are difficult

**Table 2.** The types of plants used for nut storage.

| Plant type | Number of individuals | Percentage of all individuals (%) |
|---|---|---|
| Alive tree | 108 | 71.5 |
| Dead tree | 17 | 11.3 |
| Alive liana | 19 | 12.6 |
| Dead liana | 2 | 1.3 |
| Bamboo | 5 | 3.3 |
| Total | 151 | 100 |

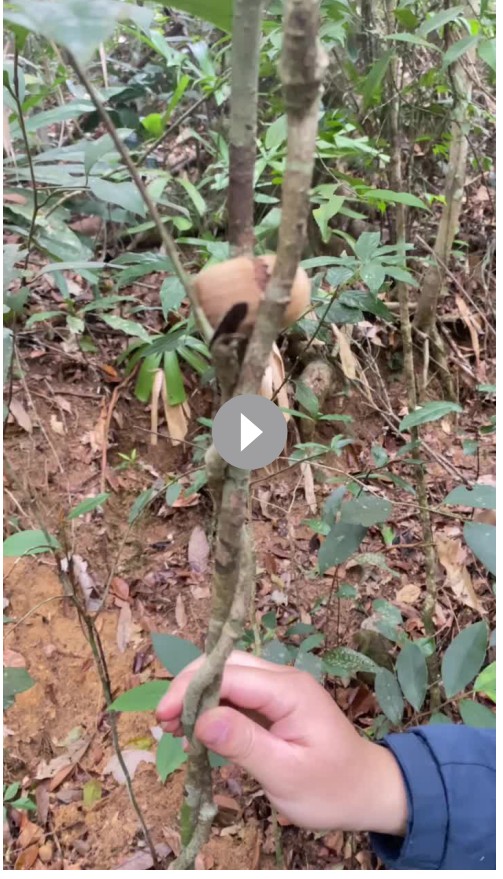

**Video 10.** Footage of shaking a liana does not dislodge nuts of *Cyclobalanopsis edithiae* stored by squirrels.
https://elifesciences.org/articles/84967/figures#video10

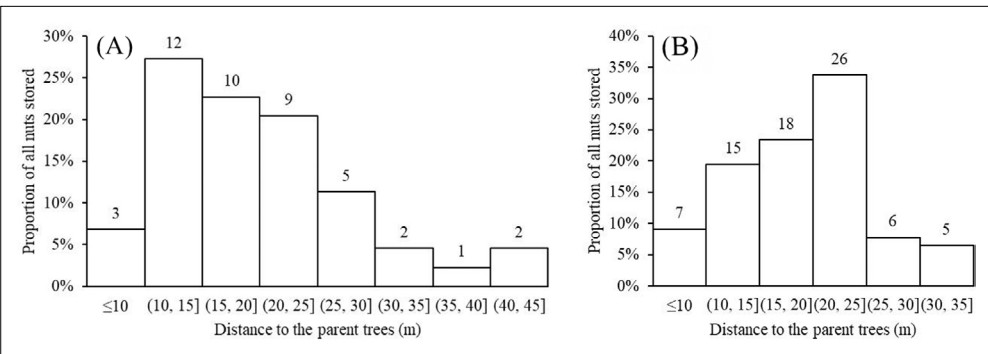

**Figure 10.** Distance from storage sites to potential parent trees for the nuts varied from 10–25 m. (**A**) *C. edithiae* nuts. (**B**) *C. patelliformis* nuts. Notes: The value on each bar is the actual number of stored nuts.

for squirrels to deal with, *Cyclobalanopsis* nuts are highly preferred as food by squirrels and other animals, although the smooth nut surfaces make them challenging to suspend on vegetation. *Cyclobalanopsis edithiae (Skan) Schottky* and *Cyclobalanopsis patelliformis* (Chun) Y. C. Hsu et H. W. Jen are the two most abundant species with naked nuts in the mountain forests of Jianfengling (*Xu et al., 2015*; *Table 3*). Both are in fruit from October to December, just before the coolest month (January) in Hainan.

**Table 3.** Main Fagaceae species found in a 60 ha plot in the Jianfengling forest.

| Species | Abundance |
|---|---|
| *Castanopsis carlesii* (Hemsley) Hayata | 3269 |
| *Castanopsis fissa* (Champion ex Bentham) Rehder & E. H. Wilson | 2803 |
| *Castanopsis jianfenglingensis* Duanmu | 2297 |
| *Castanopsis tonkinensis* Seemen | 953 |
| *Castanopsis ledongensis* C. C. Huang & Y. T. Chang | 335 |
| *Castanopsis fabri* Hance | 113 |
| *Castanopsis hystrix* J. D. Hooker & Thomson ex A. de Candolle | 35 |
| ***Cyclobalanopsis edithiae* (Skan) Schottky** | **1645** |
| ***Cyclobalanopsis patelliformis* (Chun) Y. C. Hsu & H. W. Jen** | **1207** |
| *Cyclobalanopsis phanera* (Chun) Y. C. Hsu & H. W. Jen | 886 |
| *Cyclobalanopsis fleuryi* (Hickel & A. Camus) Chun ex Q. F. Zheng | 568 |
| *Cyclobalanopsis neglecta* Schottky | 392 |
| *Cyclobalanopsis blakei* (Skan) Schottky | 279 |
| *Cyclobalanopsis hui* (Chun) Chun ex Y. C. Hsu & H. W. Jen | 220 |
| *Lithocarpus longipedicellatus* (Hickel & A. Camus) A. Camus | 2842 |
| *Lithocarpus pseudovestitus* A. Camus | 2427 |
| *Lithocarpus fenzelianus* A. Camus | 1751 |
| *Lithocarpus amygdalifolius* (Skan) Hayata | 1360 |
| *Lithocarpus handelianus* A. Camus | 1046 |
| *Lithocarpus fenestratus* (Roxburgh) Rehder | 323 |
| *Lithocarpus howii* Chun | 130 |
| *Lithocarpus hancei* (Benth.) Rehd. | 71 |

### Field investigation

During work in the Jianfengling forest, we discovered *Cyclobalanopsis* nuts with surface grooves tucked into the Y-shaped crotches of twigs on understory plants (*Figure 2*). The regular grooves showed signs of having been chewed by some unknown animal(s) (*Figure 3*), perhaps in order to increase the friction between nuts and plant twigs so as to fix the nuts securely in place. Thus, we conducted a systematic field investigation from January to May 2022 to discover the animals involved and to study their nut caching behavior in more detail.

We first made a systematic search of ca. 5.5 ha of forest for grooved nuts suspended in vegetation to seek relationships between this phenomenon and various plant types. We used plastic tags to mark all plants discovered bearing grooved nuts. The species identity, diameter at breast height (DBH), geographical position, and elevation of plants found with suspended grooved nuts were recorded. We also measured the diameter and angles of the two twigs where the nuts were fixed. Identities of trees with DBH ≥20 cm within a ca. 20 m radius of each stored nut were recorded. Finally, we measured the distance between the storage site and the nearest tree where the nuts could have been produced. Additionally, we recorded the species, weight, diameter, and height of each plant for all suspended nuts discovered. Number, depth, and position of the surface grooves were also noted for each nut. Because precise measurement was impossible, the depth of the carved grooves was measured as a categorical variable classified in the following three groups: shallow (ca. 0–0.15 mm), medium (ca. 0.15–0.30 mm), and deep (ca. 0.30–0.45 mm). We also recorded whether each nut was fresh, eaten by insects, or infected by fungi; fresh nuts are green but become wrinkled and black with age. All nuts were photographed at the time of discovery.

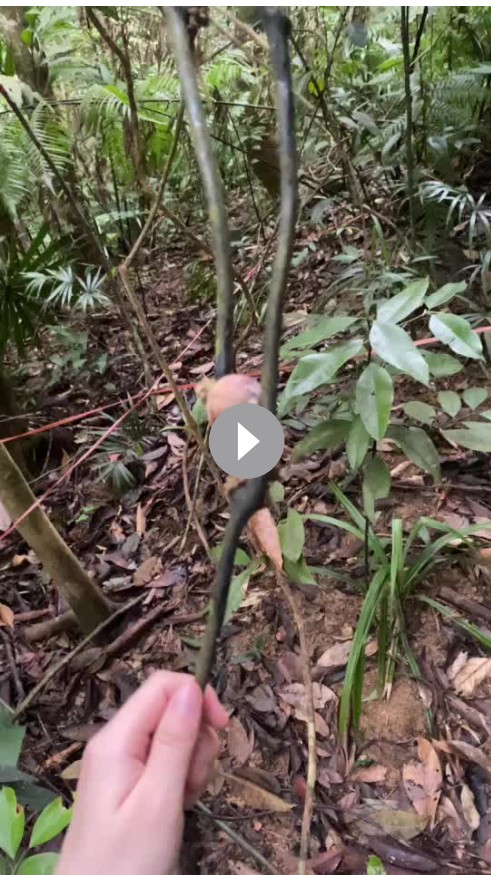

**Video 13.** Footage of shaking a liana does not dislodge nuts of *Cyclobalanopsis patelliformis* stored by squirrels.
https://elifesciences.org/articles/84967/figures#video13

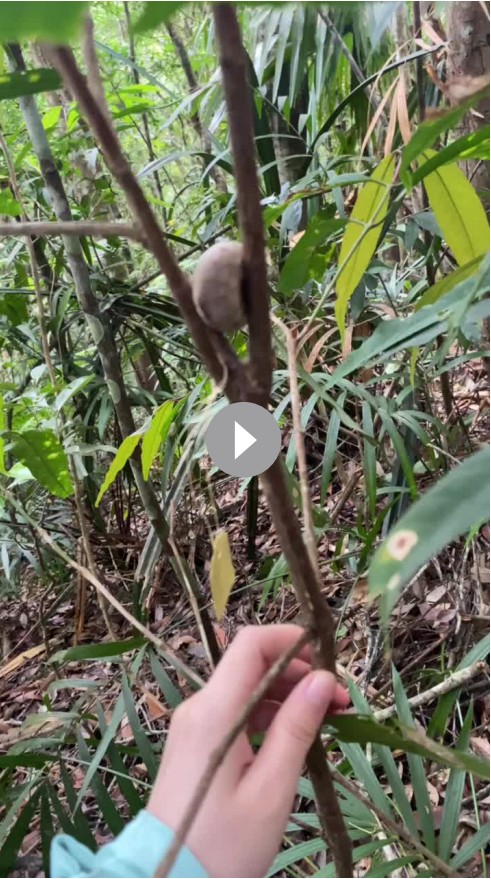

**Video 14.** Footage of shaking a sapling does not dislodge nuts of *Cyclobalanopsis patelliformis* stored by squirrels.
https://elifesciences.org/articles/84967/figures#video14

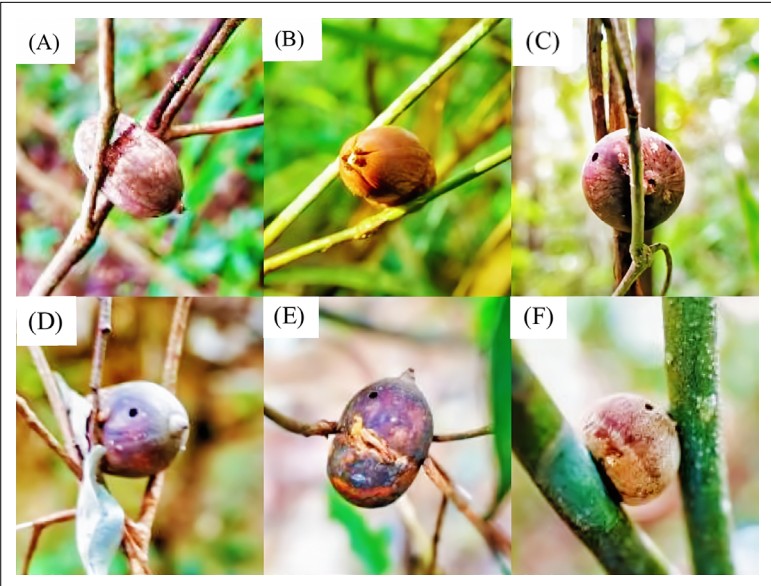

**Figure 11.** After long (e.g. >ca. 365 days) storage, nuts become not fresh. (**A**) Dried nuts, (**B**) Germinated, or (**C–E**) Destroyed by insects.

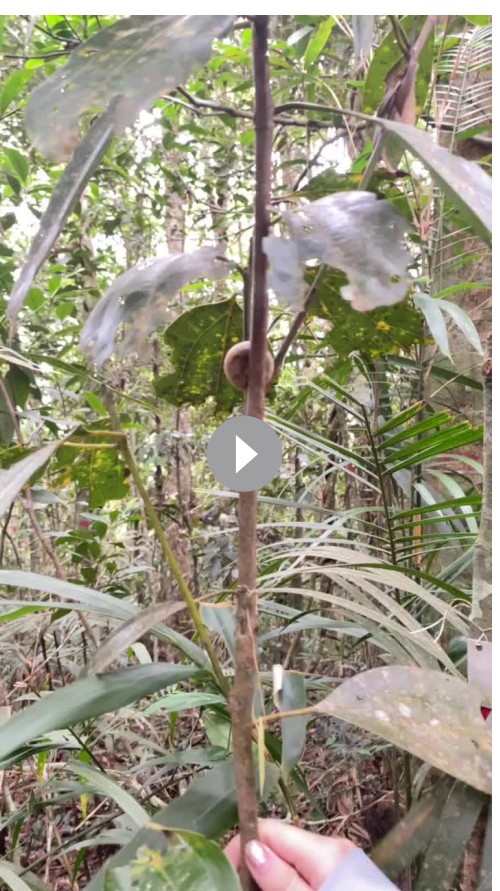

**Video 15.** Footage of shaking a sapling does not dislodge nuts of *Cyclobalanopsis patelliformis* stored by squirrels.

https://elifesciences.org/articles/84967/figures#video15

The first search for cached nuts was carried out on January 15, 2022. We re-surveyed the site 44 days later on February 28, using the same protocols described above, to learn whether the nuts recorded previously were still where we initially found them, and to find any new nuts that had been stored in the area. A third similar survey was made 61 days later on April 30.

Because we initially knew neither the identity of the animals that stored the nuts nor how nuts were stored and retrieved, we set up 32 motion-activated infrared cameras (22 WildINSights 20MP 1080 P HD Trail Cameras and 10 WildINSights 5MP 960 P HD Trail Color Cameras) around the stored nuts to monitor animal activities that might be related to nut storage or consumption. These cameras were positioned to view both typical nuts that we found and their surroundings. In general, the distance between a camera and a focal nut was 0.5–1 m. Animals filmed as being associated with the nuts were subsequently identified to species by experts using the resulting pictures and videos. Especially, for *Video 9*, we merged several photos and a video successively taken by an infrared camera in 30 s near the 64 ha permanent plot in Jianfengling, Hainan on January 23, 2023.

## Statistical analyses

We described the distribution twig angles which were used to fix the nuts and twig diameters as histograms. Standard *t*-tests were used to assess the significance of differences between paired

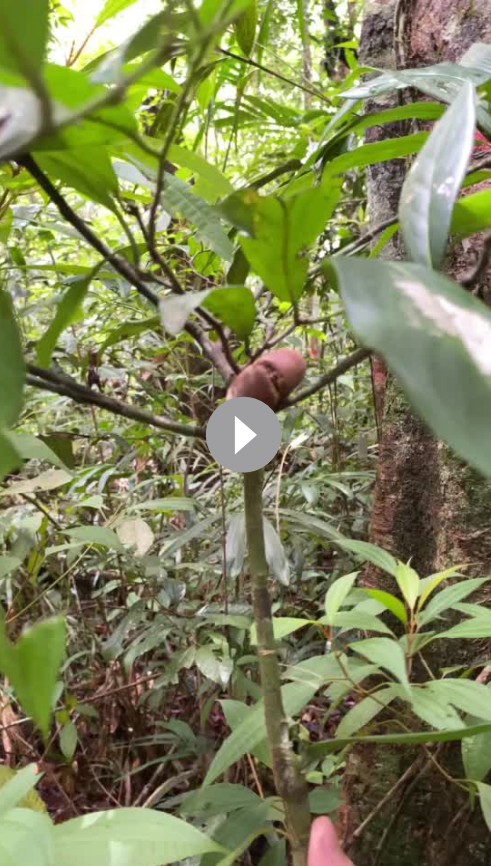

**Video 11.** Footage of shaking a sapling does not dislodge nuts of *Cyclobalanopsis edithiae* stored by squirrels.

https://elifesciences.org/articles/84967/figures#video11

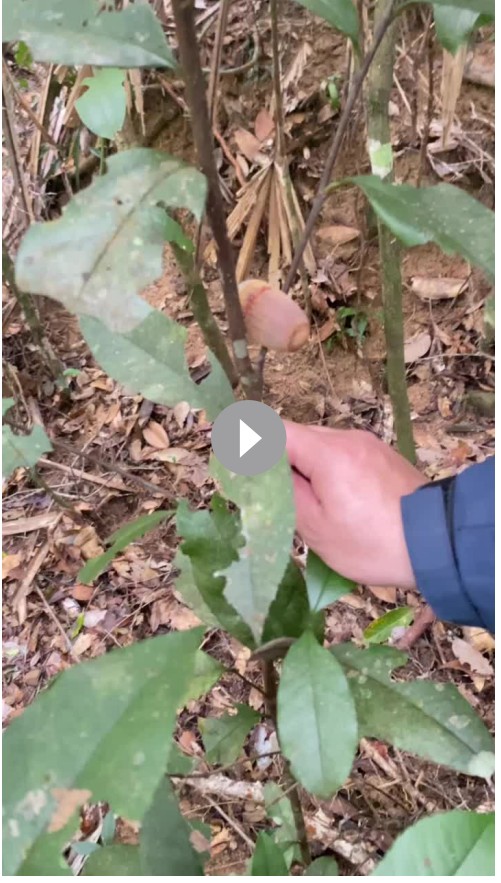

**Video 12.** Footage of shaking a sapling does not dislodge nuts of *Cyclobalanopsis edithiae* stored by squirrels.

https://elifesciences.org/articles/84967/figures#video12

sets of variables, and linear regression was used to assess relationships between the sizes of the two twigs that constituted each nut storage site. Histograms were also drawn to describe variations in the sizes and location of storage plants and carved nuts, including the width of grooves carved by squirrels, DBH and height of plants with stored nuts, number of carved grooves of each nut, and distance of stored nuts to the nearest fruiting tree of that species.

We also used t-tests to study whether the oblate nuts stored on living trees and shrubs had significantly more carved shallow scattered grooves than those stored on dead trees and lianas, and whether grooves on the ellipsoid nuts of *C. edithiae* were deeper than those on the oblate nuts of *C. patelliformis*.

## Acknowledgements

We thank Wenhao Qin, Chuanwen Yu, Fenglin Huang, and Tao Zhang for helping to collect the field data. We also appreciate suggestions from Yi Ding from Chinese Academy of Forestry that improved the manuscript, and assistance from Fang Liu from the Chinese Academy of Forestry, and Qiang Zhang from the Institute of Zoology, Guangdong Academy of Sciences in identifying the animals. Science and Technology Basic Work project from Ministry of Science and Technology of the People's Republic of China (2019FY101607).

# Additional information

### Funding

| Funder | Grant reference number | Author |
| --- | --- | --- |
| Science and Technology Basic Work Project from Ministry of Science and Technology of the People's Republic of China | 2019FY101607 | Han Xu |

The funders had no role in study design, data collection and interpretation, or the decision to submit the work for publication.

### Author contributions

Han Xu, Conceptualization, Resources, Data curation, Formal analysis, Funding acquisition, Investigation, Methodology, Writing – original draft, Project administration, Writing – review and editing; Lian Xia, Formal analysis, Investigation, Writing – original draft; John R Spence, Writing – original draft, Writing – review and editing; Mingxian Lin, Chunyang Lu, Tushou Luo, Investigation, Writing – original draft; Yanpeng Li, Investigation, Methodology, Writing – original draft; Jie Chen, Writing – original draft; Yide Li, Data curation, Writing – original draft; Suqin Fang, Conceptualization, Data curation, Formal analysis, Supervision, Methodology, Writing – original draft, Writing – review and editing

### Author ORCIDs

Han Xu ⓘ http://orcid.org/0000-0002-1085-3344
Suqin Fang ⓘ http://orcid.org/0000-0002-1324-4640

### Decision letter and Author response

Decision letter https://doi.org/10.7554/eLife.84967.sa1
Author response https://doi.org/10.7554/eLife.84967.sa2

# Additional files

### Supplementary files
• MDAR checklist
• Supplementary file 1. The plants used to store nuts and their growth form.

### Data availability
All data are available in the main text or the supplementary files.

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
