## [Editor Report]

This report of nut modification and storage in flying squirrels provides new insights into food caching behaviour in wild animals. Although further direct evidence is needed to corroborate some of the findings, the current study provides valuable documentation of an interesting behaviour that should motivate further observational and experimental research.

---

## [Decision Letter]

**Decision letter after peer review:**

Thank you for submitting your article "Smart squirrels use a mortise-tenon structure to fix nuts on understory twigs" for consideration by *eLife*.

Your article has been reviewed by three peer reviewers, including Ammie K Kalan as the Reviewing Editor and Reviewer #1, and the evaluation has been overseen by Christian Rutz as Senior Editor. The following individual involved in the review of your submission has agreed to reveal their identity: Pizza Chow (Reviewer #3).

The reviewers have discussed their reviews with one another, and the Reviewing Editor has drafted this decision letter to help you prepare a revised submission.

Essential revisions:

All three reviewers found this work interesting and novel, but agreed that the claims need to be significantly tempered given the data at hand. When revising the article, please pay attention to the detailed comments raised in the three reports, which are appended below, but ensure you address the following three major concerns:

1) Since no squirrels were actually observed creating the grooves, the whole article needs to be refocused on the recovered artefacts and observed behaviours (caching etc.). This will require a new title and substantial text revision. It should be made clear throughout that the authors have no data at present to show that flying squirrels actually made those grooves (although we would welcome any new data to support this claim, such as incisor matching mentioned by Reviewer #2). Ideally, this would be added to a section titled "Limitations of the study".

2) Related to the point above, all speculation about the evolution of this behaviour in flying squirrels and the potential underlying cognitive mechanisms should be removed from the article, since the data are simply too preliminary at present to address these points.

3) If possible, the authors should consider a more robust analysis of nut retention in the trees, with and without grooves, or a comparison of how nuts fare on the ground versus in the trees, to support their arguments (see also comments by Reviewer #3).

There is also scope for polishing the presentation, and we recommend a thorough language check.

Finally, please note that *eLife* has adopted the STRANGE framework, to help improve reporting standards and reproducibility in animal behaviour research. In your revision, please consider scope for sampling biases and potential limitations to the generalisability of your findings:

https://reviewer.elifesciences.org/author-guide/journal-policies

https://doi.org/10.1038/d41586-020-01751-5

*Reviewer #1 (Recommendations for the authors):*

I really enjoyed reading this paper and I commend the authors on the great detail with which they have recorded the behaviour. I have a few suggestions to make that I think would help to improve clarity and support to some of the text as it currently stands.

line 72-73: this sentence does not make sense, please rephrase.

line 96-97: expand upon what "actively associated" means here, i.e., just briefly state some of the behaviours observed since this is key.

You never see the squirrels actually making the grooves but it is assumed, so I think you need to clearly state this to the reader.

Can you tell us how many unique individuals were seen in your videos? at least a minimum number?

line 129: after forest, do you have a reference you can cite?

line 163: we hypothesize, rather than guess… this is also an example of a sentence needing rephrasing due to grammar issues (see last point).

line 165: I think you could clarify what learning ability you are talking about. At the moment this sentence is confusing because it seems like you are stating this is potentially tool use? It would be good to be more clear what you mean here and perhaps to also clearly state that this isn't tool use, but again to focus on or better explain what aspects of clever cognition this behaviour is evidence of.

line 168: why animal cranial zootomy? In general I could not follow your thinking in this sentence.

line 196: reference for sentence ending with 88%?

line 224: what do you mean by dominant trees?

Figure 2 caption: A) one groove (?) – the word groove is missing I think.

Figure 3: although I find the angles very interesting I think the text fails to make clear why the angle might matter. Could you include in the text for example a clear explanation of why twig angles 25-40 degrees are most frequent for this behaviour?

Figure 4 caption: I think maybe missing some words between lines 403 and 404 or else not clear

*Reviewer #2 (Recommendations for the authors):*

A bit more development of the evidence for cache loss through decomposition ('hoard rot' as it is sometimes referred) or germination and the environmental correlates of that would be helpful earlier on for the reader to understand the observations you present.

Overall the paper would benefit from some copy editing for clarity, as some sentences are a bit difficult to follow in their current form.

I have two concerns about attributing the caching to the squirrels that could be resolved by the following:

1. Direct observation of the squirrels performing this behavior.

2. Incisor matching – do the grooves carved out of the nuts match the width of the typical width of the two incisors of these species?

3. Consideration of the following: why would an arboreal species, such as a flying squirrel, cache nuts only on the lower branches of these plants, rather than higher up? That these behaviors only appear to be at eye-level to humans does make me question why squirrels would limit themselves to this height as well.

Line specific suggestions:

Line 25 – the use of 'evolutionary challenge' – I would argue it is more an ecological challenge.

Line 29 – specify what nation/region Hainan Island is in.

Line 33 – the mortise and tenon joint has a long history outside China as well; it may be more recognizable to readers to briefly articulate what the joint is used for/what it looks like rather than specify a nationality.

Line 52 – not clear what species of nuts are being discussed here, so the comment that they are in clusters doesn't immediately register. Clustering is perhaps a more species-specific phenomenon (most of the nuts I am familiar with in a temperate zone do not cluster), so this requires more explanation here.

Line 56 – again, not sure this is necessarily an evolutionary challenge but an ecological one.

Line 58 – Write out examples here. The Hunt et al. paper doesn't seem applicable here as it focuses on learning in locomotion, rather than cognitive food handling decisions. I would also avoid the use of the term 'smart' in discussing animal cognition; it is to me more of an anthropomorphic term and isn't necessary to describe the behaviour observed here, but I acknowledge this may be more of a personal preference.

Line 54 – 'most' nuts – would like to see some data to back this up, or at least examples of species that are commonly harvested by squirrels and therefore relevant here.

Line 59 – is there evidence that squirrels use tools elsewhere?

Line 68 – I would modify this – the behavior hasn't been reported elsewhere (have you checked databases of nature observations to be sure?)

Line 69 – Perhaps not to facilitate the physical lodging of a nut into a crevice, but other species of squirrels do indeed handle nuts prior to caching which improves the success rate of storage (see for example Fox 1982, Steele et al. 2001, Xiao et al. 2010).

Supplementary Figure 1 would be helpful to have in the main text.

Figure 1 – I think providing a visual comparison to the type of joint you are likening these carvings to would be helpful here, because these nut carvings appear quite different than what I would have expected given what my understanding of the mortise-tenon joint is.

Line 132 – what is the criterion for "fresh"? – OK, I see now in the methods further down.

Line 133 – how were nuts marked in the first survey to enable these conclusions?

Paragraph at Line 163 – this is very speculative; which given that this is more of a natural history observation, is fine, but it could be much more developed in order to put this study into context and to demonstrate its value. Evolution of learned behaviors that enhance survival and reproduction is a rich topic in animal behavior, and evolution of food caching behaviors in particular has received significant attention. Feeding innovations and their relation to behavioral evolution are important (as you note, tool use in corvids and parrots are excellent examples), and perhaps describing these observations against that background with citations to the literature would help make sense of this story.

Line 170 – This is late in the paper to introduce the fact that there are nine species in this region – this should be moved up.

Line 173 – have they evolved this behavior? Or is it learned or innovated? Evolution is a bold claim in the absence of familial data including fitness metrics and given how tightly clustered in time and space these observations are.

Line 180 – this is also a 'just-so' fallacy; you observed the persistence of these nuts over a very short time and did not present controls of what happens to these same species of nuts should they be stored on the ground or in the leaf litter. My experience in the tropical rain forest leads me to believe that this behavior is highly likely to have that effect of cache protection, but just because it makes logical sense does not make it data.

*Reviewer #3 (Recommendations for the authors):*

The authors described and documented a nut-storing behaviour of two species of flying squirrels that is not previously mentioned in the literature. The authors found that the two species of flying squirrels made grooves in nuts so that they can be fixed between twigs of different species of plants and shrubs. The authors conclude that the squirrels of these populations are evolved or learn to do so, as well as placed such behaviour in tool-use context and claim that they are 'smart'.

Overall, I enjoyed reading the manuscript, and knowing more about squirrels' behaviour. I also find the work is important, because there isn't proper documents stating this nut grooving behaviour in squirrels. This will spark an interest of understanding the causes and consequences of this nut grooving behaviour, which advances different fields such as animal behaviour and cognition, behaviour ecology, evolution and ecology.

Despite the strengths of the study, I do have several suggestions to improve the manuscript:

1) One of the goals of the study is to show that these squirrels are smart. Now I am not saying that they are not, but we need proper evidence, systematic investigations, and analyses to establish this claim. For now, these claims are not entirely supported by analyses or proper measurements; the definition of 'smart' and relating nut grooving behaviours to tool use is not complete. For example, no evidence indicating this behaviour involves complex cognition or cognitive process – smart, or it is related to fast learning performance – smart. All we know is the squirrels did it, but how they acquire such skills and whether it involves in complex or advance cognition is another story. With this in mind, many wordings appear to lead readers think that this is established, which is not the case.

2) Analyses – the current analyses basically is 0. Can't the authors pay a little more efforts in analyses, say using (M)ANOVA and post hoc tests to examine whether the number of as well as the depth of grooves differ in nuts that store on ground and (live and dead) trees? Or depending on the shape of a nut too? More importantly, out of how much footage did the authors capture the flying squirrels did the nut grooving behaviours? what other animal species interacted with those nuts? These are the central aims of the manuscript but are not done properly.

3) The current presentation of the manuscript is largely oriented to plants, and less about animal behaviour (definitely not involved in animal cognition). This appears to be a bit contradictory to what the authors would like to present or what they found interesting; that the grooving behaviour of seeds in flying squirrels is novel. Suggest thinking about how to reorganise this information and focus on animal behaviour

4) Figures – please present proportion or percentage on y axis, place the actual value on top of each bar in the bar graph; this will make information more presentable as well as increase the transparency of data.

[Editors' note: further revisions were suggested prior to acceptance, as described below.]

Thank you for resubmitting your article entitled "Flying squirrels use a mortise-tenon structure to fix nuts on understory twigs" for further consideration by *eLife*. Your revised article has been evaluated by Christian Rutz (Senior Editor) and a Reviewing Editor (Ammie Kalan).

The article has been much improved but there are some remaining issues that need to be addressed, as outlined below:

1) We appreciated the addition of the new videos to demonstrate the squirrels' active interaction with the nuts, but could you please address Reviewer #3's alternative explanation directly in the text and provide any further clarification regarding the carving or cracking behaviour?

2) Similarly, please modify the text to be explicit about which parts of the behaviour you have direct video evidence for and which actions are inferred and, therefore, still require further supporting evidence. This will improve transparency and help readers evaluate your findings.

3) Please address the remaining comments from the reviewers below.

Reviewer #3 (Recommendations for the authors):

Thank you very much for inviting me to review this paper again. I also thank the authors for making a great effort to revise the paper, and most of my previous suggestions/concerns have been taken into the revision. The revised paper is still interesting as it documents the grooved nut in between twigs that may be made by this species of squirrel (a behaviour that is not fully understood).

In the past version of the manuscript, one of the main weaknesses of the paper was the absence of evidence showing squirrels made grooves and fixing them in between the twigs. In this version, the authors submitted three new videos (videos 6-8), and they are useful for seeing this species of squirrel handling a nut. However, videos 6 and 7 are showing a squirrel rolling a nut and attempting to 'carve' them and Video 8 is a squirrel finding a nut and attempting to take it out between the twigs. The terminology 'carving' is debatable, as to some squirrel experts, this appears to be 'cracking' the nut shell instead of 'carving'. None of these videos really shows direct evidence as in a squirrel grooving a nut and fixing it in between the twigs. If the authors insist that the squirrels do so (which from line 92 appears to be the case), we would need videos that show a squirrel making these grooves and fitting the nut in between twigs.

---

## [Author Response]

Essential revisions:All three reviewers found this work interesting and novel, but agreed that the claims need to be significantly tempered given the data at hand. When revising the article, please pay attention to the detailed comments raised in the three reports, which are appended below, but ensure you address the following three major concerns:1) Since no squirrels were actually observed creating the grooves, the whole article needs to be refocused on the recovered artefacts and observed behaviours (caching etc.). This will require a new title and substantial text revision. It should be made clear throughout that the authors have no data at present to show that flying squirrels actually made those grooves (although we would welcome any new data to support this claim, such as incisor matching mentioned by Reviewer #2). Ideally, this would be added to a section titled "Limitations of the study".

We have included three additional videos to demonstrate that squirrels of both species rotate and carve the nuts to create the grooves. These new videos also show that squirrels can fit the nuts between twigs by carving the nuts. We think that these direct observations clearly support our claim, but agree that it was oversight not to included them in the first draft. See Videos 6-8.

2) Related to the point above, all speculation about the evolution of this behaviour in flying squirrels and the potential underlying cognitive mechanisms should be removed from the article, since the data are simply too preliminary at present to address these points.

The description about probable evolution and cognitive mechanisms of this behavior are removed or touched on only lightly in the context of discussion wrt to the grooving behavior itself.

3) If possible, the authors should consider a more robust analysis of nut retention in the trees, with and without grooves, or a comparison of how nuts fare on the ground versus in the trees, to support their arguments (see also comments by Reviewer #3).

As stated clearly in the text now, we did not find nuts stored on plants without grooves. Comparison of how the nuts fare on the ground versus stored on the trees is an interesting topic, which could be a very useful research project for the future. I added this possibility in the text.

There is also scope for polishing the presentation, and we recommend a thorough language check.

We have checked the whole text for language.

Finally, please note that eLife has adopted the STRANGE framework, to help improve reporting standards and reproducibility in animal behaviour research. In your revision, please consider scope for sampling biases and potential limitations to the generalisability of your findings:https://reviewer.elifesciences.org/author-guide/journal-policieshttps://doi.org/10.1038/d41586-020-01751-5

Thanks! We have checked and revised to whole paper to improve the reproducibility of this study.

Reviewer #1 (Recommendations for the authors):I really enjoyed reading this paper and I commend the authors on the great detail with which they have recorded the behaviour. I have a few suggestions to make that I think would help to improve clarity and support to some of the text as it currently stands.

Thanks!

line 72-73: this sentence does not make sense, please rephrase.

It has been revised make it clear.

line 96-97: expand upon what "actively associated" means here, i.e., just briefly state some of the behaviours observed since this is key.You never see the squirrels actually making the grooves but it is assumed, so I think you need to clearly state this to the reader.Can you tell us how many unique individuals were seen in your videos? at least a minimum number?

We stated some typical behaviors about the squirrels here.

Three new videos about the carving behaviors are added to the text, which shows the squirrels were actually making the grooves.

Totally 47 times of squirrels’ activities, including nuts carving, fixing, passing and take-off were seen in our videos.

line 129: after forest, do you have a reference you can cite?

We have not published the canopy width data, but by field observation. Therefore, no reference could be cited here. I added some estimated values and revised the words.

line 163: we hypothesize, rather than guess… this is also an example of a sentence needing rephrasing due to grammar issues (see last point).

Revised as suggested.

line 165: I think you could clarify what learning ability you are talking about. At the moment this sentence is confusing because it seems like you are stating this is potentially tool use? It would be good to be more clear what you mean here and perhaps to also clearly state that this isn't tool use, but again to focus on or better explain what aspects of clever cognition this behaviour is evidence of.

The whole paragraph is deleted and then contents about tool use would not be discussed in the paper.

line 168: why animal cranial zootomy? In general I could not follow your thinking in this sentence.

We removed these words.

line 196: reference for sentence ending with 88%?

Added as suggested.

line 224: what do you mean by dominant trees?

Revised. It is referred to trees with DBH>=20cm.

Figure 2 caption: A) one groove (?) – the word groove is missing I think.

Yes. Revised as suggested.

Figure 3: although I find the angles very interesting I think the text fails to make clear why the angle might matter. Could you include in the text for example a clear explanation of why twig angles 25-40 degrees are most frequent for this behaviour?

In L99-104, there is an explanation for why twig angles 25-40 degrees matter.

Figure 4 caption: I think maybe missing some words between lines 403 and 404 or else not clear

Is it “nuts”? It is added.

Reviewer #2 (Recommendations for the authors):A bit more development of the evidence for cache loss through decomposition ('hoard rot' as it is sometimes referred) or germination and the environmental correlates of that would be helpful earlier on for the reader to understand the observations you present.

We’ve added one sentence to deal with this suggestion. Furthermore, your perceptive comment has led us to the understanding that some very interesting experiments are possible to better understand the significance of this phenomenon. We hope to conduct them soon.

Overall the paper would benefit from some copy editing for clarity, as some sentences are a bit difficult to follow in their current form.

With sentences and words revised one by one.

I have two concerns about attributing the caching to the squirrels that could be resolved by the following:1. Direct observation of the squirrels performing this behavior.2. Incisor matching – do the grooves carved out of the nuts match the width of the typical width of the two incisors of these species?3. Consideration of the following: why would an arboreal species, such as a flying squirrel, cache nuts only on the lower branches of these plants, rather than higher up? That these behaviors only appear to be at eye-level to humans does make me question why squirrels would limit themselves to this height as well.

As above, we attach three new videos to show that squirrels of both species rotate and carve the nuts to create the grooves. The new videos also show that squirrels may re-fix the nuts between the twigs by further carving the nuts. These direct observations support our claims of caching innovation better.

Wrt to incisor matching, we also added new description in the Discussion to strengthen interpretation of the grooving behavior. See L113-114.

Line specific suggestions:Line 25 – the use of 'evolutionary challenge' – I would argue it is more an ecological challenge.

Revised as suggested.

Line 29 – specify what nation/region Hainan Island is in.

Revised.

Line 33 – the mortise and tenon joint has a long history outside China as well; it may be more recognizable to readers to briefly articulate what the joint is used for/what it looks like rather than specify a nationality.

Good point. Revised accordingly.

Line 52 – not clear what species of nuts are being discussed here, so the comment that they are in clusters doesn't immediately register. Clustering is perhaps a more species-specific phenomenon (most of the nuts I am familiar with in a temperate zone do not cluster), so this requires more explanation here.

The nut clusters are not emphasized here. We now state only that squirrels can hang some fruits on the branches. We have revised the whole sentence to meet problems identified in by the reviewers, and hope that things will be clearer now.

Line 56 – again, not sure this is necessarily an evolutionary challenge but an ecological one.

We agree that ‘ecological’ is better here and have revised accordingly.

Line 58 – Write out examples here. The Hunt et al. paper doesn't seem applicable here as it focuses on learning in locomotion, rather than cognitive food handling decisions. I would also avoid the use of the term 'smart' in discussing animal cognition; it is to me more of an anthropomorphic term and isn't necessary to describe the behaviour observed here, but I acknowledge this may be more of a personal preference.

These two sentences have been removed from the text and, as above, we understand that our initial thoughts about “smart” decision-making behavior were premature. Thus, we’ve restructured the paper so that any arguments about cognitive behavior are hypotheses developed from the data that we present.

Line 54 – 'most' nuts – would like to see some data to back this up, or at least examples of species that are commonly harvested by squirrels and therefore relevant here.

We have revised the paragraph to make the context smooth.

Line 59 – is there evidence that squirrels use tools elsewhere?

We removed the tool use description here.

The second paragraph was recognized with the first one to make the context smooth.

Line 68 – I would modify this – the behavior hasn't been reported elsewhere (have you checked databases of nature observations to be sure?)

Revised as suggested. We have checked the literature and do not find previous reports of this behavior.

Yes, this is true, and these references have been added to improve the completeness of the story. Thank you!

Line 69 – Perhaps not to facilitate the physical lodging of a nut into a crevice, but other species of squirrels do indeed handle nuts prior to caching which improves the success rate of storage (see for example Fox 1982, Steele et al. 2001, Xiao et al. 2010).

With sentences revised as suggested and new references cited. Line 79.

Supplementary Figure 1 would be helpful to have in the main text.

Removed to the main text as Figure 1. And all figure numbers are changed correspondingly.

Figure 1 – I think providing a visual comparison to the type of joint you are likening these carvings to would be helpful here, because these nut carvings appear quite different than what I would have expected given what my understanding of the mortise-tenon joint is.

Mortise-tenon joint is a convex-concave structure. We liken the twigs fixed the nuts as “tenon” which is convex structure, and the grooves on the nuts as “mortise” which formed a concave structure into nuts to receive a tenon. These sentences are added to the text. L32-34,138-139.

Line 132 – what is the criterion for "fresh"? – OK, I see now in the methods further down.

Yes, it is described in the methods.

Line 133 – how were nuts marked in the first survey to enable these conclusions?

We tagged all the plants using plastic tags. This is now mentioned in the revised methods. L252.

Paragraph at Line 163 – this is very speculative; which given that this is more of a natural history observation, is fine, but it could be much more developed in order to put this study into context and to demonstrate its value. Evolution of learned behaviors that enhance survival and reproduction is a rich topic in animal behavior, and evolution of food caching behaviors in particular has received significant attention. Feeding innovations and their relation to behavioral evolution are important (as you note, tool use in corvids and parrots are excellent examples), and perhaps describing these observations against that background with citations to the literature would help make sense of this story.

We removed the whole paragraph for the speculative contents.

Line 170 – This is late in the paper to introduce the fact that there are nine species in this region – this should be moved up.

Moved up as suggested.

Line 173 – have they evolved this behavior? Or is it learned or innovated? Evolution is a bold claim in the absence of familial data including fitness metrics and given how tightly clustered in time and space these observations are.

With sentences revised.

Line 180 – this is also a 'just-so' fallacy; you observed the persistence of these nuts over a very short time and did not present controls of what happens to these same species of nuts should they be stored on the ground or in the leaf litter. My experience in the tropical rain forest leads me to believe that this behavior is highly likely to have that effect of cache protection, but just because it makes logical sense does not make it data.

As suggested, we now mention that the protective value of the hanging-nut caches merits further study with comparison to the fates of nuts on the ground.

Reviewer #3 (Recommendations for the authors):The authors described and documented a nut-storing behaviour of two species of flying squirrels that is not previously mentioned in the literature. The authors found that the two species of flying squirrels made grooves in nuts so that they can be fixed between twigs of different species of plants and shrubs. The authors conclude that the squirrels of these populations are evolved or learn to do so, as well as placed such behaviour in tool-use context and claim that they are 'smart'.Overall, I enjoyed reading the manuscript, and knowing more about squirrels' behaviour. I also find the work is important, because there isn't proper documents stating this nut grooving behaviour in squirrels. This will spark an interest of understanding the causes and consequences of this nut grooving behaviour, which advances different fields such as animal behaviour and cognition, behaviour ecology, evolution and ecology.

Thanks!

Despite the strengths of the study, I do have several suggestions to improve the manuscript:1) One of the goals of the study is to show that these squirrels are smart. Now I am not saying that they are not, but we need proper evidence, systematic investigations, and analyses to establish this claim. For now, these claims are not entirely supported by analyses or proper measurements; the definition of 'smart' and relating nut grooving behaviours to tool use is not complete. For example, no evidence indicating this behaviour involves complex cognition or cognitive process – smart, or it is related to fast learning performance – smart. All we know is the squirrels did it, but how they acquire such skills and whether it involves in complex or advance cognition is another story. With this in mind, many wordings appear to lead readers think that this is established, which is not the case.

Yes, you are right. Our surprise and enthusiasm about discovering that these squirrels do this have likely pushed us to assume too much. Thus, we have revised the text and title to focus on the behaviors of grooving and storing nuts, and to downplay conclusions relating to cognitive or cognitive process. Once we have more evidence to show how squirrels develop such skills, and whether the behaviour involves complex or advance cognition, or fast learning performance, we will write another story.

2) Analyses – the current analyses basically is 0. Can't the authors pay a little more efforts in analyses, say using (M)ANOVA and post hoc tests to examine whether the number of as well as the depth of grooves differ in nuts that store on ground and (live and dead) trees? Or depending on the shape of a nut too? More importantly, out of how much footage did the authors capture the flying squirrels did the nut grooving behaviours? what other animal species interacted with those nuts? These are the central aims of the manuscript but are not done properly.

We have added detailed descriptions about how histograms and t-tests were applied.

A total of 47 footage clips capture flying squirrels, as is now mentioned in the first paragraph of results and discussion.

Interestingly, other larger-bodied squirrels passed by some plants with a stored nut, but they did not touch the nuts. Some mice were observed trying to take stored nuts, as is now mentioned in the text. L152.

3) The current presentation of the manuscript is largely oriented to plants, and less about animal behaviour (definitely not involved in animal cognition). This appears to be a bit contradictory to what the authors would like to present or what they found interesting; that the grooving behaviour of seeds in flying squirrels is novel. Suggest thinking about how to reorganise this information and focus on animal behaviour

We have rephrased the text to amplify the new finding of the grooving behaviors, which has been previously unrecognized in squirrels. We have revised the text to focus on the special behavior and how it changes nuts themselves, but also describe the vegetation sites associated with the suspended nuts that we discovered.

4) Figures – please present proportion or percentage on y axis, place the actual value on top of each bar in the bar graph; this will make information more presentable as well as increase the transparency of data.

All figures are revised as suggested.

[Editors' note: further revisions were suggested prior to acceptance, as described below.]

The article has been much improved but there are some remaining issues that need to be addressed, as outlined below:1) We appreciated the addition of the new videos to demonstrate the squirrels' active interaction with the nuts, but could you please address Reviewer #3's alternative explanation directly in the text and provide any further clarification regarding the carving or cracking behaviour?

We’ve added one new Video 9 that provides direct evidence and example of the process used by squirrels for fixing nuts. The video documents the behaviour of a squirrel that was holding a nut in its mouth and engaged in the process.

I think that the word ‘cracking’ does not appropriately express what was done by squirrels. We feel that use of the word ‘cracked’ would be misleading as the grooves are strictly surficial. This seems to be important as we point out below (Lines 128-130) “Nonetheless, none of the chewed grooves that we observed damaged the endosperm of the nut, and thus the squirrels seemed to minimize the potential impacts of fungi during storage.” The grooves, in our opinions are in fact “carved” carefully into the surface without breaking through to the inside of the nut. Thus, we have followed our antecedent expression of “chewing” in lines 85-88, taking the opportunity to establish that we infer that the chewing leads to “carved” surficial grooves. Clearly, when one refers to “cracking” nuts, say during the holiday season, it means to break the outside casing of the nut in order to get to the inside of the nut. Thus, we fear the use of “cracking” could be easily misleading. In an attempt to avoid any confusion, we have replaced “carving” with “chewing” until our word use had been clearly defined.

2) Similarly, please modify the text to be explicit about which parts of the behaviour you have direct video evidence for and which actions are inferred and, therefore, still require further supporting evidence. This will improve transparency and help readers evaluate your findings.

We added vdeo number after each action to refer the readers to direct video evidence for our claims in Line 88-90. These actions include chewing to carve grooves (Video 6-7), as well as fixing (Video 9) and removing nuts (Video 2, 3, 5) and visiting a storage site (Video 1, 4, 8). The sentence was also revised to clearly state that the videos provide direct evidence to support aspects of nut storage behaviour employed by these two squirrel species.

Reviewer #3 (Recommendations for the authors):Thank you very much for inviting me to review this paper again. I also thank the authors for making a great effort to revise the paper, and most of my previous suggestions/concerns have been taken into the revision. The revised paper is still interesting as it documents the grooved nut in between twigs that may be made by this species of squirrel (a behaviour that is not fully understood).In the past version of the manuscript, one of the main weaknesses of the paper was the absence of evidence showing squirrels made grooves and fixing them in between the twigs. In this version, the authors submitted three new videos (videos 6-8), and they are useful for seeing this species of squirrel handling a nut. However, videos 6 and 7 are showing a squirrel rolling a nut and attempting to 'carve' them and Video 8 is a squirrel finding a nut and attempting to take it out between the twigs. The terminology 'carving' is debatable, as to some squirrel experts, this appears to be 'cracking' the nut shell instead of 'carving'. None of these videos really shows direct evidence as in a squirrel grooving a nut and fixing it in between the twigs. If the authors insist that the squirrels do so (which from line 92 appears to be the case), we would need videos that show a squirrel making these grooves and fitting the nut in between twigs.

As above, we’ve added a new Video 9 that provides direct evidence of the process that squirrels use to fix the nuts.

After carefully considering the differences between these two words, “carving” has been mostly replaced by “chewing” throughout the paper. See above.